# Trajectory-wise Multiple Choice Learning for Dynamics Generalization in Reinforcement Learning

**Younggyo Seo**[*][†]    **Kimin Lee**[*][‡]    **Ignasi Clavera**[‡]    **Thanard Kurutach**[‡]
**Jinwoo Shin**[†]    **Pieter Abbeel**[‡]
[†]Korea Advanced Institute of Science and Technology
[‡]University of California, Berkeley

## Abstract

Model-based reinforcement learning (RL) has shown great potential in various control tasks in terms of both sample-efficiency and final performance. However, learning a generalizable dynamics model robust to changes in dynamics remains a challenge since the target transition dynamics follow a multi-modal distribution. In this paper, we present a new model-based RL algorithm, coined trajectory-wise multiple choice learning, that learns a multi-headed dynamics model for dynamics generalization. The main idea is updating the most accurate prediction head to specialize each head in certain environments with similar dynamics, i.e., clustering environments. Moreover, we incorporate context learning, which encodes dynamics-specific information from past experiences into the context latent vector, enabling the model to perform online adaptation to unseen environments. Finally, to utilize the specialized prediction heads more effectively, we propose an adaptive planning method, which selects the most accurate prediction head over a recent experience. Our method exhibits superior zero-shot generalization performance across a variety of control tasks, compared to state-of-the-art RL methods. Source code and videos are available at https://sites.google.com/view/trajectory-mcl.

## 1  Introduction

Deep reinforcement learning (RL) has exhibited wide success in solving sequential decision-making problems [23, 39, 45]. Early successful deep RL approaches had been mostly model-free, which do not require an explicit model of the environment, but instead directly learn a policy [25, 28, 38]. However, despite the strong asymptotic performance, the applications of model-free RL have largely been limited to simulated domains due to its high sample complexity. For this reason, model-based RL has been gaining considerable attention as a sample-efficient alternative, with an eye towards robotics and other physics domains.

The increased sample-efficiency of model-based RL algorithms is obtained by exploiting the structure of the problem: first the agent learns a predictive model of the environment, and then plans ahead with the learned model [1, 37, 42]. Recently, substantial progress has been made on the sample-efficiency of model-based RL algorithms [5, 7, 22, 23, 24]. However, it has been evidenced that model-based RL algorithms are not robust to changes in the dynamics [20, 30], i.e., dynamics models fail to provide accurate predictions as the transition dynamics of environments change. This makes model-based RL algorithms unreliable to be deployed into real-world environments where partially unspecified dynamics are common; for instance, a deployed robot might not know a priori various features of the terrain it has to navigate.

---

[*]Equal Contribution. Correspondence to {younggyo.seo@kaist.ac.kr, kiminlee@berkeley.edu}

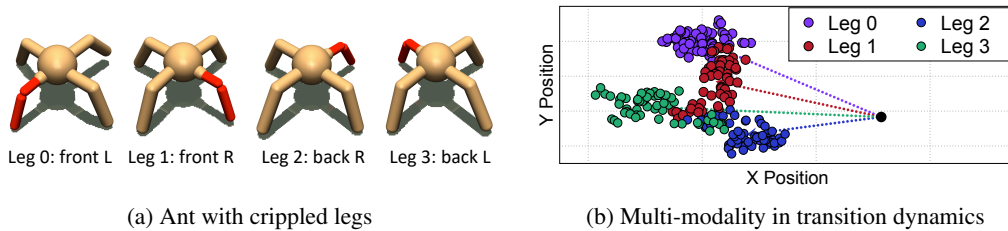

(a) Ant with crippled legs

(b) Multi-modality in transition dynamics

Figure 1: (a) Examples from ant robots with crippled legs. (b) We visualize the next $(x, y)$ positions of robots obtained by applying the same action with random noise to robots at the initial start position but with a different crippled leg.

As a motivating example, we visualize the next states obtained by crippling one of the legs of an ant robot (see Figure 1a). Figure 1b shows that the target transition dynamics follow a multi-modal distribution, where each mode corresponds to each leg of a robot, even though the original environment has deterministic transition dynamics. This implies that a model-based RL algorithm that can approximate the multi-modal distribution is required to develop a reliable and robust agent against changes in the dynamics. Several algorithms have been proposed to tackle this problem, e.g., learning contextual information to capture local dynamics [20], fine-tuning model parameters for fast adaptation [30]. These algorithms, however, are limited in that they do not explicitly learn dynamics models that can approximate the multi-modal distribution of transition dynamics.

**Contribution.** In this paper, we present a new model-based RL algorithm, coined trajectory-wise multiple choice learning (T-MCL), that can approximate the multi-modal distribution of transition dynamics in an unsupervised manner. To this end, we introduce a novel loss function, trajectory-wise oracle loss, for learning a multi-headed dynamics model where each prediction head specializes in different environments (see Figure 2a). By updating the most accurate prediction head over a trajectory segment (see Figure 2b), we discover that specialized prediction heads emerge automatically. Namely, our method can effectively cluster environments without any prior knowledge of environments. To further enable the model to perform online adaptation to unseen environments, we also incorporate context learning, which encodes dynamics-specific information from past experiences into the context latent vector and provides it as an additional input to prediction heads (see Figure 2a). Finally, to utilize the specialized prediction heads more effectively, we propose adaptive planning that selects actions using the most accurate prediction head over a recent experience, which can be interpreted as finding the nearest cluster to the current environment (see Figure 2c).

We demonstrate the effectiveness of T-MCL on various control tasks from OpenAI Gym [3]. For evaluation, we measure the generalization performance of model-based RL agents on unseen (yet related) environments with different transition dynamics. In our experiments, T-MCL exhibits superior generalization performance compared to existing model-based RL methods [4, 20, 30]. For example, compared to CaDM [20], a state-of-the-art model-based RL method for dynamics generalization, our method obtains 3.5x higher average return on the CrippledAnt environment.

## 2 Related work

**Model-based reinforcement learning.** By learning a forward dynamics model that approximates the transition dynamics of environments, model-based RL attains a superior sample-efficiency. Such a learned dynamics model can be used as a simulator for model-free RL methods [16, 18, 40], providing a prior or additional features to a policy [9, 47], or planning ahead to select actions by predicting the future consequences of actions [1, 22, 42]. A major challenge in model-based RL is to learn accurate dynamics models that can provide correct future predictions. To this end, numerous methods thus have been proposed, including ensembles [4] and latent dynamics models [14, 15, 37]. While these methods have made significant progress even in complex domains [15, 37], dynamics models still struggle to provide accurate predictions on unseen environments [20, 30].

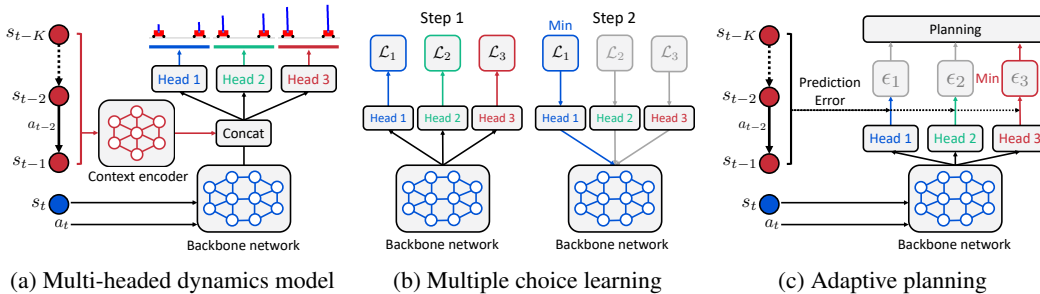

(a) Multi-headed dynamics model     (b) Multiple choice learning     (c) Adaptive planning

Figure 2: Illustrations of our framework. (a) Our dynamics model consists of multiple prediction heads conditioned on a context latent vector. (b) We train our multi-headed dynamics model with the proposed trajectory-wise oracle loss (2). (c) For planning, we select the most accurate prediction head over a recent experience.

**Multiple choice learning.** Multiple choice learning (MCL) [12, 13] is an ensemble method where the objective is to minimize an oracle loss, making at least one ensemble member predict the correct answer. By making the most accurate model optimize the loss, MCL encourages the model to produce multiple outputs of high quality. Even though several optimization algorithms [8, 12] have been proposed for MCL objectives, it is challenging to employ these for learning deep neural networks due to training complexity. To address this problem, Lee et al. [21] proposed a stochastic gradient descent based algorithm. Recently, several methods [19, 29, 43] have been proposed to further improve MCL by tackling the issue of overconfidence problem of neural networks. While most prior works on MCL have focused on supervised learning, our method applies the MCL algorithm to model-based RL.

**Dynamics generalization and adaptation in model-based RL.** Prior dynamics generalization methods have aimed to either encode inductive biases into the architecture [36] or to learn contextual information that captures the local dynamics [20]. Notably, Lee et al. [20] introduced a context encoder that captures dynamics-specific information of environments, and improved the generalization ability by providing a context latent vector as additional inputs. Our method further improves this method by combining multiple choice learning and context learning.

For dynamics adaptation, several meta-learning based methods have been studied [30, 31, 35]. Recently, Nagabandi et al. [30] proposed a model-based meta-RL method that adapts to recent experiences either by updating model parameters via a small number of gradient updates [11] or by updating hidden representations of a recurrent model [10]. Our method differs from this method, in that we do not fine-tune the model parameters to adapt to new environments at evaluation time.

## 3 Problem setup

We consider the standard RL framework where an agent optimizes a specified reward function through interacting with an environment. Formally, we formulate our problem as a discrete-time Markov decision process (MDP) [41], which is defined as a tuple $(\mathcal{S}, \mathcal{A}, p, r, \gamma, \rho_0)$. Here, $\mathcal{S}$ is the state space, $\mathcal{A}$ is the action space, $p\left(s'|s,a\right)$ is the transition dynamics, $r\left(s,a\right)$ is the reward function, $\rho_0$ is the initial state distribution, and $\gamma \in [0,1)$ is the discount factor. The goal of RL is to obtain a policy, mapping from states to actions, that maximizes the expected return defined as the total accumulated reward. We tackle this problem in the context of model-based RL by learning a forward dynamics model $f$, which approximates the transition dynamics $p\left(s'|s,a\right)$. Then, dynamics model $f$ is used to provide training data for a policy or predict the future consequences of actions for planning.

In order to address the problem of generalization, we further consider the distribution of MDPs, where the transition dynamics $p_c\left(s'|s,a\right)$ varies according to a context $c$. For instance, a robot agent's transition dynamics may change when some of its parts malfunction due to unexpected damages. Our goal is to learn a generalizable forward dynamics model that is robust to such dynamics changes, i.e., approximating the multi-modal distribution of transition dynamics $p\left(s'|s,a\right) = \int_c p\left(c\right) p_c\left(s'|s,a\right)$. Specifically, given a set of training environments with contexts sampled from $p_{\texttt{train}}(c)$, we aim to learn a forward dynamics model that can produce accurate predictions for both training environments and test environments with unseen (but related) contexts sampled from $p_{\texttt{test}}(c)$.

---

**Algorithm 1** Trajectory-wise MCL (T-MCL).

---

Initialize parameters of backbone network $\theta$, prediction heads $\{\theta_h^{\mathtt{head}}\}_{h=1}^{H}$, context encoder $\phi$.
Initialize dataset $\mathcal{B} \leftarrow \emptyset$.
**for** each iteration **do**
    Sample $c \sim p_{\mathtt{seen}}(c)$.                         // ENVIRONMENT INTERACTION
    **for** $t = 1$ **to** TaskHorizon **do**
        Get context latent vector $z_t = g\left(\tau_{t,K}^{\mathtt{P}}; \phi\right)$ and select the best prediction head $h^*$ from (3)
        Collect $\{(s_t, a_t, s_{t+1}, r_t, \tau_{t,K}^{\mathtt{P}})\}$ from environment with transition dynamics $p_c$ using $h^*$.
        Update $\mathcal{B} \leftarrow \mathcal{B} \cup \{(s_t, a_t, s_{t+1}, r_t, \tau_{t,K}^{\mathtt{P}})\}$.
    **end for**
    Initialize $L_{\mathtt{tot}} \leftarrow 0$                    // DYNAMICS AND CONTEXT LEARNING
    Sample $\{\tau_{t_j,K}^{P}, \tau_{t_j,M}^{F}\}_{j=1}^{B} \sim \mathcal{B}$
    **for** $j = 1$ **to** $B$ **do**
        **for** $h = 1$ **to** $H$ **do**
            Compute the loss of the $h$-th prediction head:

$$L_j^h \leftarrow -\frac{1}{M} \sum_{i=t_j}^{t_j+M-1} \log f\left(s_{i+1} \mid b(s_i, a_i; \theta), g(\tau_{i,K}^{\mathtt{P}}; \phi); \theta_h^{\mathtt{head}}\right)$$

        **end for**
        Find $h^* = \mathrm{argmin}_{h \in [H]} L_j^h$ and update $L_{\mathtt{tot}} \leftarrow L_{\mathtt{tot}} + L_j^{h^*}$
    **end for**
    Update $\theta, \phi, \{\theta_h^{\mathtt{head}}\}_{h=1}^{H} \leftarrow \nabla_{\theta, \phi, \{\theta_h^{\mathtt{head}}\}_{h=1}^{H}} L_{\mathtt{tot}}$
**end for**

---

# 4 Trajectory-wise multiple choice learning

In this section, we propose a trajectory-wise multiple choice learning (T-MCL) that learns a multi-headed dynamics model for dynamics generalization. We first present a trajectory-wise oracle loss for making each prediction head specialize in different environments, and then introduce a context-conditional prediction head to further improve generalization. Finally, we propose an adaptive planning method that generates actions by planning under the most accurate prediction head over a recent experience for planning.

## 4.1 Trajectory-wise oracle loss for multi-headed dynamics model

To approximate the multi-modal distribution of transition dynamics, we introduce a multi-headed dynamics model $\{f\left(s_{t+1}|b(s_t, a_t; \theta); \theta_h^{\mathtt{head}}\right)\}_{h=1}^{H}$ that consists of a backbone network $b$ parameterized by $\theta$ and $H$ prediction heads parameterized by $\{\theta_h^{\mathtt{head}}\}_{h=1}^{H}$ (see Figure 2a). To make each prediction head specialize in different environments, we propose a trajectory-wise oracle defined as follows:

$$\mathcal{L}^{\mathtt{T\text{-}MCL}} = \mathbb{E}_{\tau_{t,M}^{\mathtt{F}} \sim \mathcal{B}} \left[ \min_{h \in [H]} -\frac{1}{M} \sum_{i=t}^{t+M-1} \log f\left(s_{i+1} \mid b(s_i, a_i; \theta); \theta_h^{\mathtt{head}}\right) \right], \tag{1}$$

where $[H]$ is the set $\{1, \cdots, H\}$, $\tau_{t,M}^{\mathtt{F}} = (s_t, a_t, \cdots, s_{t+M-1}, a_{t+M-1}, s_{t+M})$ denotes a trajectory segment of size $M$, and $\mathcal{B} = \{\tau_{t,M}^{\mathtt{F}}\}$ is the training dataset. The proposed loss is designed to only update the most accurate prediction head over each trajectory segment for specialization (see Figure 2b). By considering the accumulated prediction error over trajectory segments, the proposed oracle loss can assign trajectories from different transition dynamics to different transition heads more distinctively (see Figure 4 for supporting experimental results). Namely, our method clusters environments in an unsupervised manner. We also remark that the shared backbone network learns common features across all environments, which provides several advantages, such as improving sample-efficiency and reducing computational costs.

### 4.2 Context-conditional multi-headed dynamics model

To further enable the dynamics model to perform online adaptation to unseen environments, we introduce a context encoder $g$ parameterized by $\phi$, which produces a latent vector $g\left(\tau_{t,K}^{\text{P}};\phi\right)$ given $K$ past transitions $(s_{t-K}, a_{t-K}, \cdots, s_{t-1}, a_{t-1})$. This context encoder operates under the assumption that the true context of the underlying MDP can be captured from recent experiences [20, 30, 34, 48]. Using this context encoder, we propose to learn a context-conditional multi-headed dynamics model optimized by minimizing the following oracle loss:

$$\mathcal{L}_{\text{context}}^{\text{T-MCL}} = \mathbb{E}_{(\tau_{t,M}^{\text{F}}, \tau_{t,K}^{\text{P}}) \sim \mathcal{B}} \left[ \min_{h \in [H]} -\frac{1}{M} \sum_{i=t}^{t+M-1} \log f\left(s_{i+1} \mid b(s_i, a_i; \theta), g(\tau_{i,K}^{\text{P}}; \phi); \theta_h^{\text{head}}\right) \right]. \quad (2)$$

We remark that the dynamics generalization of T-MCL can be enhanced by incorporating the contextual information into the dynamics model for enabling its online adaptation. To extract more meaningful contextual information, we also utilize various auxiliary prediction losses proposed in Lee et al. [20] (see the supplementary material for more details).

### 4.3 Adaptive planning

Once a multi-headed dynamics model is learned, it can be used for selecting actions by planning. Since the performance of planning depends on the quality of predictions, it is important to select the prediction head specialized in the current environment for planning. To this end, following the idea of Narendra & Balakrishnan [32], we propose an adaptive planning method that selects the most accurate prediction head over a recent experience (see Figure 2c). Formally, given $N$ past transitions, we select the prediction head $h^*$ as follows:

$$\underset{h \in [H]}{\arg\min} \sum_{i=t-N}^{t-2} \ell\left(s_{i+1}, f\left(b(s_i, a_i), g(\tau_{i,K}^{\text{P}}; \phi); \theta_h^{\text{head}}\right)\right), \quad (3)$$

where $\ell$ is the mean squared error function. One can note that this planning method corresponds to finding the nearest cluster to the current environment. We empirically show that this adaptive planning significantly improves the performance by selecting the prediction head specialized in a specific environment (see Figure 6b for supporting experimental results).

## 5 Experiments

In this section, we designed our experiments to answer the following questions:

- How does our method compare to existing model-based RL methods and state-of-the-art model-free meta-RL method (see Figure 3)?
- Can prediction heads be specialized for a certain subset of training environments with similar dynamics (see Figure 4 and Figure 5)?
- Is the multi-headed architecture useful for dynamics generalization of other model-based RL methods (see Figure 6a)?
- Does adaptive planning improve generalization performance (see Figure 6b)?
- Can T-MCL extract meaningful contextual information from complex environments (see Figure 6c and Figure 6d)?

### 5.1 Setups

**Environments.**   We demonstrate the effectiveness of our proposed method on classic control problems (i.e., CartPoleSwingUp and Pendulum) from OpenAI Gym [3] and simulated robotic continuous tasks (i.e., Hopper, SlimHumanoid, HalfCheetah, and CrippledAnt) from MuJoCo physics engine [44]. To evaluate the generalization performance, we designed environments to follow a multi-modal distribution by changing the environment parameters (e.g., length and mass) similar to Packer et al. [33] and Zhou et al. [48]. We use the two predefined discrete set of environment parameters for training and test environments, where parameters for test environments are outside the range of

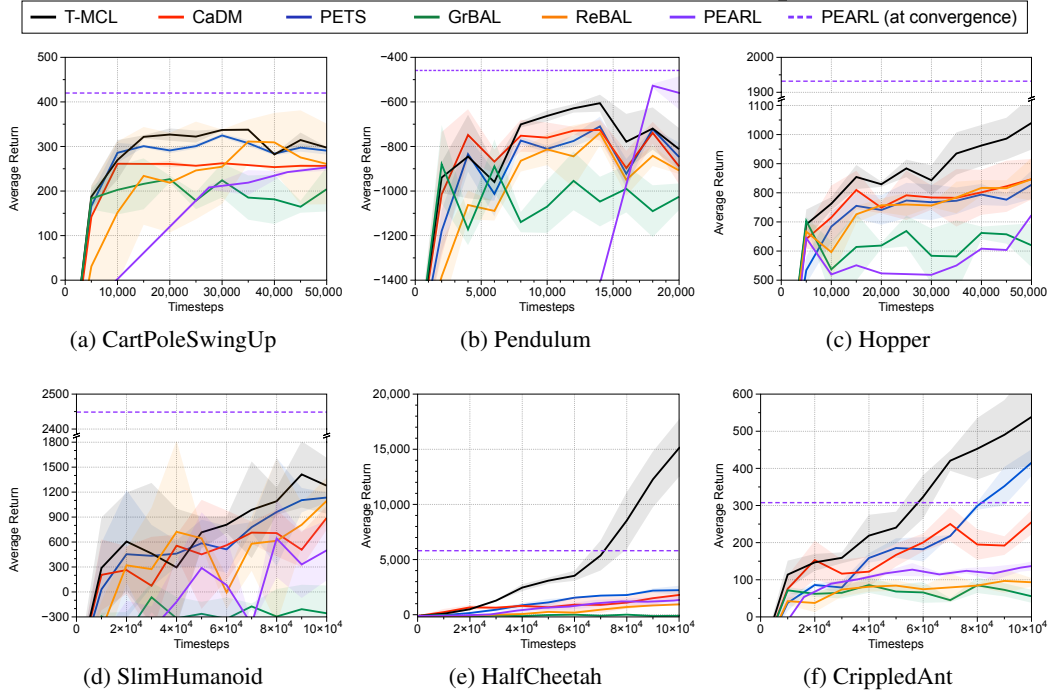

Figure 3: The average returns of trained dynamics models on unseen environments. Dotted lines indicate performance at convergence. The solid lines and shaded regions represent mean and standard deviation, respectively, across three runs.

training parameters. Then, we learn a dynamics model on environments whose transition dynamics are characterized by the environment parameters randomly sampled before the episode starts. For evaluation, we report the performance of a trained dynamics model on unseen environments whose environment parameters are randomly sampled from the test parameter set. Similar to prior works [4, 30, 46], we assume that the reward function of environments is known, i.e., ground-truth rewards at the predicted states are available for planning. For all our experiments, we report the mean and standard deviation across three runs. We provide more details in the supplementary material.

**Planning.** We use a model predictive control (MPC) [27] to select actions based on the learned dynamics model. Specifically, we use the cross entropy method (CEM) [6] to optimize action sequences by iteratively re-sampling action sequences near the best performing action sequences from the last iteration.

**Implementation details of T-MCL.** For all experiments, we use an ensemble of multi-headed dynamics models that are independently optimized with the trajectory-wise oracle loss. We reduce modeling errors by training multiple dynamics models [4]. To construct a trajectory segment $\tau_{t,M}^{\mathrm{F}}$ in (1), we use $M$ transitions randomly sampled from a trajectory instead of consecutive transitions $(s_t, a_t, \cdots, s_{t+M})$. We empirically found that this stabilizes the training, by breaking the temporal correlations of the training data. We also remark that the same hyperparameters are used for all experiments except Pendulum which has a short task horizon. We provide more details in the supplementary material.

**Baselines.** To evaluate the performance of our method, we consider following model-based and model-free RL methods:

- Probabilistic ensemble dynamics model (PETS) [4]: an ensemble of probabilistic dynamics models that captures the uncertainty in modeling and planning. PETS employs ensembles of single-headed dynamics models optimized to cover all training environments, while T-MCL employs ensembles of multi-headed dynamics models specialized to a subset of environments.

| Mass | Head 1 | Head 2 | Head 3 |
|---|---|---|---|
| 0.25 | 95.3 | 4.7 | 0.0 |
| 0.50 | 0.0 | 100.0 | 0.0 |
| 1.50 | 0.0 | 0.2 | 99.8 |
| 2.50 | 0.0 | 0.0 | 100.0 |

(a) CartPoleSwingUp

| Length | Head 1 | Head 2 | Head 3 |
|---|---|---|---|
| 0.50 | 100.0 | 0.0 | 0.0 |
| 0.75 | 0.0 | 0.0 | 100.0 |
| 1.0 | 10.1 | 89.9 | 0.0 |
| 1.25 | 0.1 | 99.9 | 0.0 |

(b) Pendulum

| Mass | Head 1 | Head 2 | Head 3 |
|---|---|---|---|
| 0.25 | 100.0 | 0.0 | 0.0 |
| 0.50 | 73.6 | 26.4 | 0.0 |
| 1.50 | 2.0 | 2.1 | 95.9 |
| 2.50 | 0.7 | 86.9 | 12.4 |

(c) HalfCheetah

Figure 4: Fraction of training trajectories assigned to each prediction head optimized by the proposed trajectory-wise oracle loss (2) on (a) CartPoleSwingUp, (b) Pendulum, and (c) HalfCheetah. Number in the $(i, j)$-th cell of each table denotes the fraction of trajectories with $i$-th environment parameter, i.e., mass and length, assigned to $j$-th head of a multi-headed dynamics model.

| Mass | Head 1 | Head 2 | Head 3 |
|---|---|---|---|
| 0.50 | 46.3 | 29.6 | 24.1 |
| 0.75 | 51.1 | 28.0 | 20.9 |
| 1.0 | 50.2 | 28.3 | 21.5 |
| 1.25 | 52.9 | 27.0 | 20.1 |
| 1.50 | 50.6 | 27.7 | 21.8 |

(a) Multiple choice learning (MCL)

| Mass | Head 1 | Head 2 | Head 3 |
|---|---|---|---|
| 0.50 | 83.3 | 11.0 | 5.6 |
| 0.75 | 57.9 | 31.3 | 10.9 |
| 1.0 | 12.5 | 63.5 | 24.0 |
| 1.25 | 5.2 | 30.5 | 64.3 |
| 1.50 | 5.3 | 16.8 | 77.9 |

(b) Trajectory-wise MCL (T-MCL)

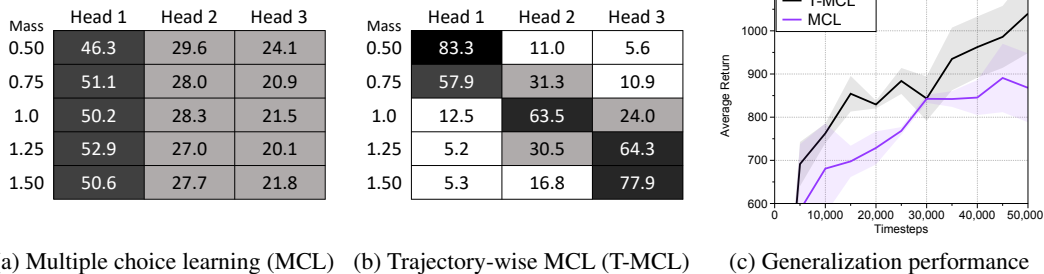

(c) Generalization performance

Figure 5: Fraction of training trajectories assigned to each prediction head optimized by (a) MCL and (b) T-MCL on Hopper environments. Number in the $(i, j)$-th cell of each table denotes the fraction of trajectories with $i$-th environment parameter, i.e., mass, assigned to $j$-th head of a multi-headed dynamics model. (c) Generalization performance of dynamics models trained with MCL and T-MCL on unseen Hopper environments.

- Model-based meta-learning methods (ReBAL and GrBAL) [30]: model-based meta-RL methods capable of adapting to a recent trajectory segment, by updating a hidden state with a recurrent model (ReBAL), or by updating model parameters with gradient updates (GrBAL).

- Context-aware dynamics model (CaDM) [20]: a dynamics model conditioned on a context latent vector that captures dynamics-specific information from past experiences. For all experiments, we use the combined version of CaDM and PETS.

- Model-free meta-learning method (PEARL) [34]: a context-conditional policy that conducts adaptation by inferring the context using trajectories from the test environment. A comparison with this method evaluates the benefit of model-based RL methods, e.g., sample-efficiency.

## 5.2 Comparative evaluation on control tasks

Figure 3 shows the generalization performances of our method and baseline methods on unseen environments (see the supplementary material for training curve plots). Our method significantly outperforms all model-based RL baselines in all environments. In particular, T-MCL achieves the average return of 19280.1 on HalfCheetah environments while that of PETS is 2223.9. This result demonstrates that our method is more effective for dynamics generalization, compared to the independent ensemble of dynamics models. On the other hand, model-based meta-RL methods (ReBAL and GrBAL) do not exhibit significant performance gain over PETS, which shows the difficulty of adapting a dynamics model to unseen environments via meta-learning. We also remark that CaDM does not consistently improve over PETS, due to the difficulty in context learning with a discrete number of training environments. We observed that T-MCL sometimes reaches the performance of PEARL or even outperforms it in terms of both sample-efficiency and asymptotic performance. This result demonstrates the effectiveness of our method for dynamics generalization, especially given that PEARL adapts to test environments by collecting trajectories at evaluation time.

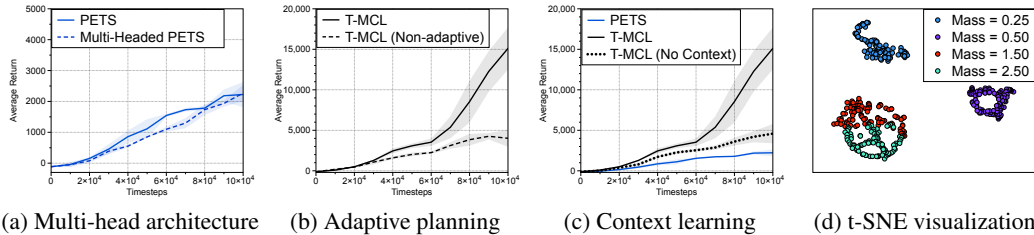

| (a) Multi-head architecture | (b) Adaptive planning | (c) Context learning | (d) t-SNE visualization |

Figure 6: (a) Generalization performance of PETS and Multi-Headed PETS on unseen HalfCheetah environments. (b) We compare the generalization performance of adaptive planning to non-adaptive planning on unseen HalfCheetah environments. (c) Generalization performance of trained dynamics models on unseen HalfCheetah environments. One can observe that T-MCL still outperforms PETS without context learning, but this results in a significant performance drop. (d) t-SNE visualization of hidden features of context-conditional multi-headed dynamics model on HalfCheetah environments.

## 5.3 Analysis

**Specialization.** To investigate the ability of our method to learn specialized prediction heads, we visualize how training trajectories are assigned to each head in Figure 4. One can observe that trajectories are distinctively assigned to prediction heads, while trajectories from environments with similar transition dynamics are assigned to the same prediction head. For example, we discover that the transition dynamics of Pendulum with length 1.0 and 1.25 are more similar to each other than Pendulum with other lengths (see the supplementary material for supporting figures), which implies that our method can cluster environments in an unsupervised manner.

**Effects of trajectory-wise loss.** To further investigate the effectiveness of trajectory-wise oracle loss, we compare our method to MCL, where we consider only a single transition for selecting the model to optimize, i.e., $M = 1$ in (1). Figure 5a and Figure 5b show that training trajectories are more distinctively assigned to each head when we use T-MCL, which implies that trajectory-wise loss is indeed important for learning specialized prediction heads. Also, as shown in Figure 5c, this leads to superior generalization performance over the dynamics model trained with MCL, showing that learning specialized prediction heads improves the generalization performance.

**Effects of multi-headed dynamics model.** We also analyze the isolated effect of employing multi-headed architecture on the generalization performance. To this end, we train the multi-headed version of PETS, i.e., ensemble of multi-headed dynamics models without trajectory-wise oracle loss, context learning, and adaptive planning. Figure 6a shows that multi-headed PETS does not improve the performance of vanilla PETS on HalfCheetah environments, which demonstrates the importance of training with trajectory-wise oracle loss and adaptively selecting the most accurate prediction head for achieving superior generalization performance of our method.

**Effects of adaptive planning.** We investigate the importance of selecting the specialized prediction head adaptively. Specifically, we compare the performance of employing the proposed adaptive planning method to the performance of employing non-adaptive planning, i.e., planning with the average predictions of prediction heads. As shown in Figure 6b, the gain due to adaptive planning is significant, which confirms that proposed adaptive planning is important.

**Effects of context learning.** We examine our choice of integrating context learning by comparing the performance of a context-conditional multi-headed dynamics model to the performance of a multi-headed dynamics model. As shown in Figure 6c, removing context learning scheme from the T-MCL results in steep performance degradation, which demonstrates the importance of incorporating contextual information. However, we remark that the T-MCL still outperforms PETS without context learning scheme. Also, we visualize the hidden features of a context-conditional multi-headed dynamics model on HalfCheetah environments using t-SNE [26] in Figure 6d. One can observe that features from the environments with different transition dynamics are separated in the embedding space, which implies that our method indeed learns meaningful contextual information.

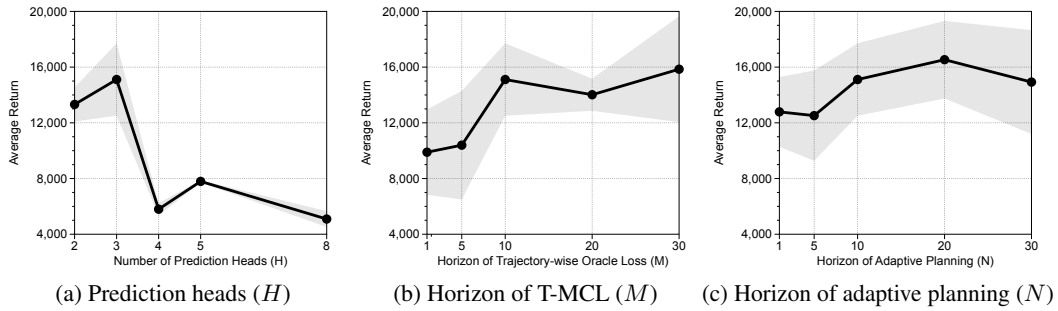

(a) Prediction heads ($H$)    (b) Horizon of T-MCL ($M$)    (c) Horizon of adaptive planning ($N$)

Figure 7: Performance of dynamics models trained with T-MCL on unseen HalfCheetah environments with varying (a) the number of prediction heads, (b) horizon of trajectory-wise oracle loss, and (c) horizon of adaptive planning.

**Effects of hyperparameters.**    Finally, we investigate how hyperparameters affect the performance of T-MCL. Specifically, we consider three hyperparameters, i.e., $H \in \{2, 3, 4, 5, 8\}$ for the number of prediction heads in (2), $M \in \{1, 5, 10, 20, 30\}$ for the horizon of trajectory-wise oracle loss in (2), and $N \in \{1, 5, 10, 20, 30\}$ for the horizon of adaptive planning in (3). Figure 7a shows that $H = 3$ achieves the best performance because three prediction heads are enough to capture the multi-modality of the training environments in our setting. When $H > 3$, the performance decreases because trajectories from similar dynamics are split into multiple heads. Figure 7b and Figure 7c show that our method is robust to the horizons $M, N$, and considering more transitions can further improve the performance. We provide results for all environments in the supplementary material.

## 6    Conclusion

In this work, we present trajectory-wise multiple choice learning, a new model-based RL algorithm that learns a multi-headed dynamics model for dynamics generalization. Our method consists of three key ingredients: (a) trajectory-wise oracle loss for multi-headed dynamics model, (b) context-conditional multi-headed dynamics model, and (c) adaptive planning. We show that our method can capture the multi-modal nature of environments in an unsupervised manner, and outperform existing model-based RL methods. Overall, we believe our approach would further strengthen the understanding of dynamics generalization and could be useful to other relevant topics such as model-based policy optimization methods [15, 16].

## Broader Impact

While deep reinforcement learning (RL) has been successful in a range of challenging domains, it still suffers from a lack of generalization ability to unexpected changes in surrounding environmental factors [20, 30]. This failure of autonomous agents to generalize across diverse environments is one of the major reasoning behind the objection to real-world deployment of RL agents. To tackle this problem, in this paper, we focus on developing more robust and generalizable RL algorithm, which could improve the applicability of deep RL to various real-world applications, such as robotics manipulation [17] and package delivery [2]. Such advances in the robustness of RL algorithm could contribute to improved productivity of society via the safe and efficient utilization of autonomous agents in a diverse range of industries.

Unfortunately, however, we could also foresee the negative long-term consequences of deploying autonomous systems in the real-world. For example, autonomous agents could be abused by specifying harmful objectives such as autonomous weapons. While such malicious usage of autonomous agents was available long before the advent of RL algorithms, developing an RL algorithm for dynamics generalization may accelerate the real-world deployment of such malicious robots, e.g., autonomous drones loaded with explosives, by making them more robust to changing dynamics or defense systems. We would like to recommend the researchers to recognize this potential misuse as we further improve RL systems.

## Acknowledgments and Disclosure of Funding

We thank Junsu Kim, Seunghyun Lee, Jongjin Park, Sihyun Yu, and our anonymous reviewers for feedback and discussions. This research is supported in part by ONR PECASE N000141612723, Tencent, Berkeley Deep Drive, Institute of Information & communications Technology Planning & Evaluation (IITP) grant funded by the Korea government (MSIT) (No.2019-0-00075, Artificial Intelligence Graduate School Program (KAIST)), and Engineering Research Center Program through the National Research Foundation of Korea (NRF) funded by the Korean Government MSIT (NRF-2018R1A5A1059921).

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
