[Supplementary Material]

# Supplementary Material

## A    Details on experimental setups

### A.1    Environments

Figure 8: Visualization of multi-modal distribution in (a) CartPoleSwingUp, (b) Pendulum, (c) Hopper, (d) SlimHumanoid, (e) HalfCheetah, and (f) CrippledAnt environments. We first collect trajectory from the default environment (black colored transitions in figures) and visualize the next states obtained by applying the same action to the same state with different environment parameters. One can observe that transition dynamics follow multi-modal distributions.

**CartPoleSwingUp.**    For CartPoleSwingUp environments, we use open source implementation of CartPoleSwingUp[2], which is the modified version of original CartPole environments from OpenAI Gym [3]. The objective of CartPoleSwingUp is to swing up the pole by moving a cart and keep the pole upright within 500 time steps. For our experiments, we modified the mass of cart and pole within the set of $\{0.25, 0.5, 1.5, 2.5\}$ and evaluated the generalization performance in unseen environments with a mass of $\{0.1, 0.15, 2.75, 3.0\}$. We visualize the transitions in Figure 8a.

**Pendulum.**    We use the Pendulum environments from the OpenAI Gym [3]. The objective of Pendulum is to swing up the pole and keep the pole upright within 200 time steps. We modified the length of pendulum within the set of $\{0.5, 0.75, 1.0, 1.25\}$ and evaluated the generalization performance in unseen environments with a length of $\{0.25, 0.375, 1.5, 1.75\}$. We visualize the transitions in Figure 8b.

**Hopper.**    We use the Hopper environments from MuJoCo physics engine [44]. The objective of Hopper is to move forward as fast as possible while minimizing the action cost within 500 time steps. We modified the mass of a hopper robot within the set of $\{0.5, 0.75, 1.0, 1.25, 1.5\}$ and evaluated the generalization performance in unseen environments with a mass of $\{0.25, 0.375, 1.75, 2.0\}$. We visualize the transitions in Figure 8c.

**SlimHumanoid.** We use the modified version of Humanoid environments from MuJoCo physics engine [44][3]. The objective of SlimHumanoid is to move forward as fast as possible while minimizing the action cost within 1000 time steps. We modified the mass of a humanoid robot within the set of $\{0.8, 0.9, 1.0, 1.15, 1.25\}$ and evaluated the generalization performance in unseen environments with a mass of $\{0.6, 0.7, 1.5, 1.6\}$. We visualize the transitions in Figure 8d.

**HalfCheetah.** We use the HalfCheetah environments from MuJoCo physics engine [44]. The objective of HalfCheetah is to move forward as fast as possible while minimizing the action cost within 1000 time steps. We modified the mass of a halfcheetah robot within the set of $\{0.25, 0.5, 1.5, 2.5\}$ and evaluated the generalization performance in unseen environments with a mass of $\{0.1, 0.15, 2.75, 3.0\}$. We visualize the transitions in Figure 8e.

**CrippledAnt.** We use the modified version of Ant environments from MuJoCo physics engine [44][4]. The objective of CrippledAnt is to move forward as fast as possible while minimizing the action cost within 1000 time steps. We randomly crippled one of three legs in a ant robot and evaluated the generalization performance by crippling remaining one leg of the ant robot. We visualize the transitions in Figure 8f.

## A.2 Training

We train dynamics models for 10 iterations for all experiments. Each iteration consists of data collection and updating model parameters. First, we collect 10 trajectories with MPC controller from environments with varying environment parameters. For planning via MPC, we use the cross entropy method with 200 candidate actions and 5 iterations to optimize action sequences with horizon 30. We also use $N = 10$ for the number of past transitions in adaptive planning (3). Then we update the model parameters with Adam optimizer of learning rate 0.001. To effectively increase the prediction ability of each prediction head before specialized prediction heads emerge, we train all prediction heads independently on all environments for initial 3 iterations, instead of training each prediction head seperately from the first iteration. Except for Pendulum environments, we update model parameters for 50 epochs. For Pendulum environments, we update model parameters for 5 epochs due to the short horizon length of the task.

## A.3 Architecture

Following Chua et al. [4], our backbone network is modeled as multi-layer perceptrons (MLPs) with 4 hidden layers of 200 units each and Swish activations. Each prediction head is modeled as Gaussian, in which the mean and variance are parameterized by a single linear layer which takes the output vector of the backbone network as an input. We use $H = 3$ for the number of prediction heads and $M = 10$ for the size of trajectory segment in (2). We remark that hyperparameters $H$ and $M$ are selected based on trajectory assignments, i.e., how distinctively trajectories are assigned to each prediction head. We use an ensemble of 5 multi-headed dynamics models that are independently trained on entire training environments and 20 particles for trajectory sampling. Note that we do not use bootstrap models. For context-conditional dynamics model, following Lee et al. [20], a context encoder is modeled as MLPs with 3 hidden layers that produce a 10-dimensional vector. Then, this context vector is concatenated with the output vector of a backbone network.

## A.4 Auxiliary prediction losses for context learning

Here, we provide a more detailed explanation of how we implemented various prediction losses proposed in Lee et al. [20]. These losses are proposed to force context latent vector to be useful for predicting both (a) forward dynamics and (b) backward dynamics. Specifically, we compute the proposed forward and backward prediction losses by substituting log-likelihood loss for proposed trajectory-wise oracle loss (2). In order to further remove the computational cost of optimization, we first compute the assignments, i.e., $h^*$ for each transition, through the entire dataset and optimize the forward and backward prediction losses with mini-batches.

# B  Learning curves

Figure 9: The average returns of trained dynamics models on training environments. Dotted lines indicate performance at convergence. The solid lines and shaded regions represent mean and standard deviation, respectively, across three runs.

# C  Effects of multi-headed dynamics model

Figure 10: Generalization performance of PETS and Multi-Headed PETS on unseen (a) Cart-PoleSwingUp, (b) Pendulum, (c) Hopper, (d) SlimHumanoid, (e) HalfCheetah, and (f) CrippledAnt environments. The solid lines and shaded regions represent mean and standard deviation, respectively, across three runs.

# D   Effects of adaptive planning

Figure 11: Generalization performance of employing adaptive planning and non-adaptive planning on unseen (a) CartPoleSwingUp, (b) Pendulum, (c) Hopper, (d) SlimHumanoid, (e) HalfCheetah, and (f) CrippledAnt environments. The solid lines and shaded regions represent mean and standard deviation, respectively, across three runs.

# E   Effects of context learning

Figure 12: Generalization performance of trained dynamics models on unseen (a) CartPoleSwingUp, (b) Pendulum, (c) Hopper, (d) SlimHumanoid, (e) HalfCheetah, and (f) CrippledAnt environments. The solid lines and shaded regions represent mean and standard deviation, respectively, across three runs.

# F  Effects of trajectory-wise loss

Figure 13: Generalization performance of dynamics models trained with MCL and T-MCL on unseen (a) CartPoleSwingUp, (b) Pendulum, (c) Hopper, (d) SlimHumanoid, (e) HalfCheetah, and (f) CrippledAnt environments. The solid lines and shaded regions represent mean and standard deviation, respectively, across three runs.

# G  Effects of hyperparameters

## G.1  Number of prediction heads

Figure 14: Generalization performance of dynamics models trained with MCL on unseen (a) CartPoleSwingUp, (b) Pendulum, (c) Hopper, (d) SlimHumanoid, (e) HalfCheetah, and (f) CrippledAnt environments with varying number of prediction heads. The solid lines and shaded regions represent mean and standard deviation, respectively, across three runs.

## G.2 Horizon of trajectory-wise oracle loss

(a) CartPoleSwingUp

(b) Pendulum

(c) Hopper

(d) SlimHumanoid

(e) HalfCheetah

(f) CrippledAnt

Figure 15: Generalization performance of dynamics models trained with MCL on unseen (a) Cart-PoleSwingUp, (b) Pendulum, (c) Hopper, (d) SlimHumanoid, (e) HalfCheetah, and (f) CrippledAnt environments with varying horizon of trajectory-wise oracle loss. The solid lines and shaded regions represent mean and standard deviation, respectively, across three runs.

## G.3 Horizon of adaptive planning

(a) CartPoleSwingUp

(b) Pendulum

(c) Hopper

(d) SlimHumanoid

(e) HalfCheetah

(f) CrippledAnt

Figure 16: Generalization performance of dynamics models trained with MCL on unseen (a) Cart-PoleSwingUp, (b) Pendulum, (c) Hopper, (d) SlimHumanoid, (e) HalfCheetah, and (f) CrippledAnt environments with varying horizon of adaptive planning. The solid lines and shaded regions represent mean and standard deviation, respectively, across three runs.

## Footnotes

[2]We use implementation available at `https://github.com/angelolovatto/gym-cartpole-swingup`

[3]We use implementation available at `https://github.com/WilsonWangTHU/mbbl`

[4]We use implementation available at `https://github.com/iclavera/learning_to_adapt`