[Reviews · NeurIPS 2020]

Review 1

Summary and Contributions: This paper addresses the problem of a reinforcement learning agent that has been trained in a variety of related environments and now faces a novel (still related) environment. The goal is to help the agent quickly adapt to the new dynamics. The main idea here is to leverage Multiple-Choice Learning, in which an ensemble is trained with the goal that at least one member of the ensemble should make the correct prediction (this is in contrast with other ensemble frameworks in which predictions are made by averaging predictions or by majority vote). In this case MCL is implemented by adding multiple prediction heads to a dynamics model. The heads are encouraged to specialize in different dynamics by training only the most accurate head in any given example. Experiments demonstrate in simulated robotics domains that T-MCL can adapt to unseen dynamics more readily than existing approaches and ablation studies show the importance of various components of the architecture.

Strengths: - The problem of developing agents that are robust to small changes in dynamics is important for making RL approaches more applicable in messy real-world settings. - Presents a novel (as far as I know) application of MCL to model-based RL. - Empirical work is strong: in addition to positive "cook-off" results, investigatory experiments support hypotheses about why the approach is effective and several ablation studies investigate the importance of individual components and their integration.

Weaknesses: A weakness of the paper is that I didn't find a clear discussion of drawbacks and limitations of the approach. Under what circumstances is this approach likely to fail? What implicit assumptions are made by the design decisions? One that occurs to me is that it doesn't seem like this approach would improve transfer/adaptation to a single change in dynamics and instead relies on the agent being exposed to a variety of MDPs in order to interpolate between them. The paper would be strengthened if it, in addition to demonstrating the benefits of the approach, clearly identified problems left open and barriers left to surmount.

Correctness: I am not aware of technical errors in this work.

Clarity: For the most part I found the paper to be clearly expressed. The main exception to this occurs in the beginning. From the abstract and the introduction I honestly had a hard time nailing down what problem this paper was even addressing. The term "model-based RL" is used a fair bit, but that is not specific enough. This paper is not just addressing the RL problem with a model-based approach, but rather it goes after the multi-task learning or meta-learning or transfer learning, or whatever you want to label this with. I think that needs to be made much more clear up front. Also there is much attention given to "the multi-model nature of environments." Even after a close reading of the entire paper I am still not 100% sure I understand what this means. What distribution is multi-modal in this statement? Do all environments have this property? What causes this multi-modality and why is it a problem for existing work? Now that I have read the whole paper I can make guesses about how the authors would answer those questions, but I think it's important that these foundational points be super clear in the beginning of the paper.

Relation to Prior Work: The authors do a good job of surveying existing work that addresses adaptation to changes in dynamics in the RL literature.

Reproducibility: Yes

Additional Feedback: After author response: Thank you to the authors for this response. I feel that most of my comments have been addressed to my satisfaction. I will re-emphasize that the introduction needs to do a much better job of clearly expressing the hypothesized connection between dynamics generalization and multi-modality. The paper needs to offer more argument to connect these two and, in so doing, it needs to clearly identify the limits of this intuition (i.e. are there dynamics generalization tasks where multi-modality is not a key issue).


Review 2

Summary and Contributions: - This paper proposes to use multi-headed dynamics models to better represent the multi-modal nature of dynamics modeling - They evaluate their proposed method on a suite of classical control tasks using MPC for planning/control - Their proposed method significantly outperforms existing approaches - Their analysis shows that specialization emerges amount of multiple prediction heads

Strengths: - The proposed model is a sensible and logical combination of multi-choice learning and dynamics model learning - The proposed method is either on-par with or considerably outperforms existing approaches - The analysis and ablations of the proposed model are thorough and interesting

Weaknesses: - I would like to see an ablation on the number of heads used. - When choosing the head, should it be done per-step or per-trajectory? The experiment with M=1 somewhat answers this question but also entangles the optimization process with it.

Correctness: The work is well situated with respect to prior work

Clarity: The paper is clearly written and the figures are very nicely made and helpful. My only suggestion/nit with the writing is to remove the phrase "Due to space limitation" -- what was moved to the supplement was logical to move, no need to make excuses.

Relation to Prior Work: The work is well situated with respect to prior work. It nicely builds-upon and applies advanced in multi-choice learning to model-based RL.

Reproducibility: Yes

Additional Feedback: Post Rebuttal ---------------- I thank the authors for addressing my concerns. After discussion with the other reviewers, I have decided to maintain my rating. The ablation of the number of heads is somewhat concerning and the explanation as to why performance reduces as the number of heads increases does not explain why 4 heads is worse than 3 and 5. What I think is happening is that over specialization leads to heads that don't generalize well. You can't have perfectly specialized heads as otherwise none will generalize well. This failure mode isn't dramatic and even if it was, I don't think it should block this paper from being published, so I am noting it for completeness.


Review 3

Summary and Contributions: Paper proposes to improve the existing model-based RL methods with ensemble-based/multi-head dynamic model (e.g. PETS, PlaNet) from several angles, including a trajectory-wise loss, an auxiliary context learning task and an adaptive planning heuristic. The proposed improvement is evaluated on some transferring tasks, where the dynamics slightly changes across training and testing envs, and demonstrate some advantages over multiple baselines.

Strengths: + The paper is overall well-written and easy to follow. The authors make their major contribution pretty clear. + The proposed improvement has limited novelty but is technically sound. + Related work and counterparts are discussed and compared in a reasonable way. The technical contributions are evaluated extensively. The results are convincing.

Weaknesses: Overall I find this submission makes no major mistakes and perhaps should be considered as a borderline one, given their contributions are more like some engineering endeavor but are evaluated properly, if there is no other reviewer points out some significant issues. I only have a few comments on this paper at this point: - If my understanding is correct, I feels the horizon of traj-wise loss (M) adaptive planning (N) and number of heads (H) are major hyper-parameters in the proposed improvement, but I can't find any illustrations on how can these three be selected during their evaluations. The authors are expected to provide a clarification on how the parameter search is conducted. A comprehensive ablations on how do these parameters affect the results is also expected. - Since the proposed improvement is agnostic to the concrete MBRL algorithm, experiments with different planning backbone should be included, e.g. Dreamer (https://openreview.net/forum?id=S1lOTC4tDS), MBPO(https://arxiv.org/abs/1906.08253), etc.

Correctness: I can't find any technical issues within this paper.

Clarity: The paper is overall well-written. The contributions are made clear and easy to follow,

Relation to Prior Work: The authors do an excellent job on discussing and comparing with counterparts.

Reproducibility: Yes

Additional Feedback: Post-rebuttal ==== Thanks for the extra ablation results. Although there are still a few points are missing(error bars, more tasks, etc), it's understandable given the limited time. Please include those results (with analysis and those missing points fixed) in your final draft. Besides those above, I would encourage the authors to elaborate more on why combining the idea of ensemble learning (PETS) and multi-head model could enjoy those extrapolation advantages. Some theoretical analysis or more in-depth discussion on this could possibly deliver greater impact to our community. I've updated my score per the changes have been made so far.


Review 4

Summary and Contributions: This paper studies the problem of learning robust generalizable dynamics models for model-based reinforcement learning in environments with multi-modal dynamics. Authors present a novel model-based RL algorithm based on a multi-headed dynamics model that is a) optimized trajectory-wise, b) conditioned on prior context and c) whose heads are selected adaptively for optimal planning. The presented model outperforms state-of-the-art model-based baselines and results comparable to a model-free algorithm.

Strengths: - The problem and approach are well motivated and grounded in theory and prior work. - Three clear technical contributions are presented in the paper: a) a multi-headed dynamics model trained via a novel trajectory-wise oracle loss, b) a context encoder that conditions the multi-headed dynamics predictions on prior states and actions, and c) an adaptive planning method that selects the dynamics model head based on how well heads predicted in the current environment in the past. - The empirical evaluation is thorough, and well motivated and organized. - The approach outperforms state-of-the-art model-based baselines on classic control problems and simulated robotic continuous tasks. - Qualitative results show that all heads are indeed utilized. - Ablation studies show that each of the presented contributions helps with performance. - Authors took the time to think properly about the broader impact of their work. - The paper is well-written, well-structured and easily understandable. - This work is both significant in that it addressed changing dynamical environments and generalization across environments which is an outstanding problem for model-based methods and is to my knowledge novel to the NeurIPS community.

Weaknesses: - The approach does not seem to outperform the model-free PEARL baseline on every task at convergence. - The number of heads has been fixed to three and either a better explanation could have been given why three heads were chosen or a more detailed analysis would be necessary if three is the optimal number of heads or whether the number of heads needs to change depending on how strongly and how many physical parameters vary in the environment. - Are there any interesting qualitative results on the control tasks? It might be interesting to visualize what happens when the different heads take over control for the different tasks. E.g. how does it look like when the best head steers the ant with crippled legs vs how does it look like when the worst head steers it? (assuming it is possible to manually fix the head in your method of course)

Correctness: - Claims and methods as well as empirical methodology and results seem to be correct, at least to my knowledge.

Clarity: - The paper is very well written and organized. The introduction motivates the problem well. The related work is detailed and relevant and explains the delta in the technical approach. The introduced technical contributions are well supported by experiments that address questions about performance against baselines as well as how much each contribution contributes to the final performance in ablation studies. Results are well analyzed and summarized. Overall, a very well-rounded paper.

Relation to Prior Work: - Related work is detailed and nicely organized. - The approach is clearly delineated to prior work and technical contributions are clearly mentioned, namely authors introduce a multi-headed dynamics model for model-based reinforcement learning that is a) optimized trajectory-wise, b) conditioned on prior context and c) whose heads are selected adaptively for optimal planning.

Reproducibility: Yes

Additional Feedback: Authors have addressed all of my questions during the rebuttal by providing a hyperparameter search on the number of heads, explaining why their method some times does not outperform PEARL and ran the suggested qualitative experiment which will be included as a video. Other reviewers have not raised any major concerns that I might have missed. Thus, I remain with my recommendation to accept this paper.

[Author Response · NeurIPS 2020]

We thank all reviewers for carefully reading our paper and their valuable comments. We appreciate that our paper is recognized for several positive aspects: [**R1**, **R2**, **R4**] novel and well-motivated, [**R2**, **R3**, **R4**] clear write-up, and [**All**] extensive and strong experiments. Below are our responses to the reviewers, which we will incorporate in the final draft.

| Mass | Head 1 | Head 2 | Head 3 | Head 4 | Head 5 |
|------|--------|--------|--------|--------|--------|
| 0.25 | 0.0 | 0.0 | 0.0 | 100.0 | 0.0 |
| 0.50 | 0.1 | 0.0 | 0.0 | 0.0 | 99.9 |
| 1.50 | 56.3 | 43.7 | 0.0 | 0.0 | 0.0 |
| 2.50 | 0.0 | 44.3 | 55.7 | 0.0 | 0.0 |

(a) Prediction heads      (b) Trajectory assignment      (c) Trajectory-wise oracle loss      (d) Adaptive planning

[**R2**, **R3**, **R4**] **Effects of major hyperparameters.** We choose the number $H$ of prediction heads and the horizon $M$ of trajectory-wise oracle loss based on the trajectory assignments in training environments, i.e., how distinctively trajectories are assigned to prediction heads (see Figure 4 and 5 of the original draft and above Figure (b) for examples). Also, the horizon $N$ of adaptive planning is set to the same value as $M$ to match the horizon of trajectory-wise oracle loss and adaptive planning. We use the same hyperparameters $H$, $M$, and $N$ for all environments. We will clarify the hyperparameter setups in the final draft.

Following the suggestions of **R2**, **R3**, and **R4**, we also perform ablation studies in HalfCheetah environments by varying the major hyperparameters of our method: $H \in \{2, 3, 4, 5, 8\}$, $M \in \{1, 5, 10, 20, 30\}$, and $N \in \{1, 5, 10, 20, 30\}$. Figure (a) shows that $H = 3$ achieves the best performance because three prediction heads are enough to capture the multi-modality of the training environments in our setting. When $H > 3$, the performance decreases because trajectories from similar dynamics are split into multiple heads as shown in Figure (b). However, as pointed out by **R4**, we expect that more heads would be effective in environments with more varying environmental factors. Figure (c) and (d) show that choosing the head per-trajectory is more effective and our method is robust to change in hyperparameters $M \geq 10$ and $N \geq 10$. We will include more comprehensive results for all environments in the final draft.

[**R1**] **Limitation of our method.** Thank you very much for your pointers. As we assume that MDPs with similar dynamics will behave similarly, the effectiveness of our method would be limited if the dynamics of unseen environments are significantly different from the dynamics of training environments. As you pointed out, gain from our method may decrease if the agent is not exposed to a variety of MDPs. We will clarify these limitations in the final draft.

[**R1**] **Clarification on problem formulation.** Our work addresses the dynamics generalization problem of model-based RL methods, where learned dynamics models fail to provide accurate predictions as the transition dynamics of environments change. Thank you for your suggestion, and we will clarify this in the final draft.

[**R1**] **Clarification on multi-modal nature of environments.** We assume that the transition dynamics distribution of MDP is multi-modal, which emerges as the environmental factors (e.g., mass and length of the agent) change. We remark that capturing this property is important as environmental factors are ever-changing in real-world environments. Prior model-based RL methods that do not consider this property fail to generalize as the future prediction from dynamics models becomes inaccurate. We will clarify this in the final draft.

[**R2**] **Editorial comment.** We will remove the "Due to space limitation" phrase in the final draft.

[**R3**] **Novelty.** As **R1**, **R2**, and **R4** pointed out, we believe that we propose a novel and well-motivated combination of multiple choice learning and model-based RL, whose major components are (i) context-conditional multi-headed dynamics model, (ii) trajectory-wise oracle loss, and (iii) adaptive planning. We also verify the effectiveness of the proposed method for improving dynamics generalization via exhaustive ablation studies on various benchmarks.

[**R3**] **Extension to other model-based RL methods.** Applying our trajectory-wise multiple choice learning scheme to model-based policy optimization methods (e.g., MBPO and Dreamer) is an interesting direction, and we think our method is naturally applicable. For example, one can consider employing our method for learning specialized policies using specialized prediction heads. We leave this as future work, but will include related discussion in the final draft.

[**R3**] **Broader Impact.** We will add more discussion related to the broader impact of our work to the final draft.

[**R4**] **Comparison to PEARL.** We emphasize that our method is evaluated using the zero-shot generalization performance in test environments, while PEARL is evaluated using the few-shot adaptation performance, i.e., PEARL conducts adaptation to test environments before evaluation. Our method still achieves superior sample efficiency compared to PEARL in most environments and outperforms PEARL in HalfCheetah and CrippledAnt environments even in terms of asymptotic performance. We also remark that as in PEARL, it is possible to learn a context-conditional policy using context vectors from our method (e.g., see [1]) to further improve performance.

[**R4**] **Qualitative analysis on control tasks.** This is a very interesting question. Following your suggestion, we visualize the behavior of a cheetah agent using the prediction head specialized for low-mass environments on the HalfCheetah environment with a default body mass. Interestingly, we observe that the cheetah agent moves as *if* it has a lightweight body, i.e., moving its limbs about very fast. We will include videos in the final draft.

[1] K. Lee et al. Context-aware dynamics model for generalization in model-based RL. In *ICML*, 2020.

[Meta-Review · NeurIPS 2020]

Reviewers are in favor of acceptance after the discussion and I agree. The key novelty in this work is to apply the Multiple Choice Learning framework to model based reinforcement learning. Doing so allows for the model to learn multimodal distributions over future states and the authors provide strong empirical results. Neither dynamics learning nor MCL are novel; however, their combination is novel and demonstrated to be effective. The reviewers have left a number of useful suggestions about how to further strengthen the paper in terms of writing and experimentation and I encourage the authors to make use of this feedback.